# Introduction of a leaky stop codon as molecular tool in *Chlamydomonas reinhardtii*

Oliver D. Caspari ⓘ *

UMR7141, Institut de Biologie Physico-Chimique (CNRS/Sorbonne Université), Paris, France

* odc20@cantab.net

## Abstract

Expression of proteins in the chloroplast or mitochondria of the model green alga *Chlamydomonas reinhardtii* can be achieved by directly inserting transgenes into organellar genomes, or through nuclear expression and post-translational import. A number of tools have been developed in the literature for achieving high expression levels from the nuclear genome despite messy genomic integration and widespread silencing of transgenes. Here, recent advances in the field are combined and two systems of bicistronic expression, based on ribosome reinitiation or ribosomal skip induced by a viral 2A sequence, are compared side-by-side. Further, the small subunit of Rubisco (RBCS) was developed as a functional nuclear reporter for successful chloroplast import and restoration of photosynthesis: To be able to combine RBCS with a Venus fluorescent reporter without compromising photosynthetic activity, a leaky stop codon is introduced as a novel molecular tool that allows the simultaneous expression of functional and fluorescently tagged versions of the protein from a single construct.

**Data Availability Statement:** All relevant data are within the manuscript and its Supporting Information files.

**Funding:** ODC received financial support from (a) LabEx Dynamo: ANR-LABX-011, http://www.labexdynamo.ibpc.fr/en/ (b) the Rothschild

## Introduction

*Chlamydomonas reinhardtii* is currently the only organism known to science where transgenes can be expressed from all three eukaryotic genomes, located in nucleus, mitochondria and the chloroplast [1–3]. This genetic control forms part of a suite of molecular tools that have made the green alga an attractive model organism for research and increasingly also for biotechnology [2–4]. For research on organelles in particular, Chlamydomonas is well suited: unlike higher plant systems, Chlamydomonas can be grown heterotrophically, allowing mutants in essential photosynthetic genes in both the nuclear and chloroplast genomes to be propagated [5–7]. Likewise, when kept continuously in the light, photoautotrophy allows mutants in essential respiratory genes in the nuclear and mitochondrial genomes to be isolated [8,9]. Moreover, the only other organism where mitochondrial genomes can be transformed, *Sacharomyces cerevisiae*, lacks complex I, which has made Chlamydomonas the model of choice for research on this key respiratory complex [10,11].

In order to express proteins in the chloroplast or mitochondria, gene constructs can be introduced into organelle genomes directly via biolistic bombardment, where they will be integrated via homologous recombination [12,13]. Alternatively, transgenes can be introduced

Foundation: http://www.fedr.ibpc.fr/en/ (c) the ChloroMitoRAMP grant, co-ordinated by Yves Choquet: ANR-19-CE13-0009, https://anr.fr/fileadmin/documents/2019/Aapg-selection-prc-prce-jcjc-2019.pdf (d) annual funding from the Centre National de la Recherche Scientifique and Sorbonne Université to UMR7141. The funders had no role in study design, data collection and analysis, decision to publish, or preparation of the manuscript.

**Competing interests:** The author has declared that no competing interests exist.

into the nuclear genome and gene products targeted to organelles post-translationally. This may be preferable in certain circumstances. While a number of antibiotic resistance selectable marker genes are available for chloroplast and nuclear transformation [13,14], only *ble* has been established for mitochondria [15]. The presence of multiple copies of the genome in both organelles makes multiple rounds of selection necessary in an attempt to reach homoplasmy, compared to a single selection step for the haploid nuclear genome [2,13]. Finally, there is no *in situ* transcriptional regulation of organelle genomes, meaning conditional expression of transgenes requires control via nuclear factors [16].

Nuclear expression of transgenes is not without its own challenges, which a lot of work has gone into addressing [14]. One problem is that Chlamydomonas appears to possess a powerful suite of gene silencing mechanisms which need to be circumvented [14]. Adjusted codon usage [17], use of the *HSP70A-RBCS2* (AR) fusion promoter [18] and inclusion of introns in the construct [19] help to avoid silencing and boost expression. *RBCS2* intron 1 in particular is a popular choice, as it has been shown to contain an enhancer element [20].

Another challenge is the absence of homologous recombination in the nuclear genome. Instead, transformation cassettes are integrated essentially at random [21], leading to large variation in expression levels due to the influence of genomic location on transcription rates [22]. Furthermore, endonucleolytic cleavage of the transformation cassette is common, which can lead to insertion of truncated fragments [6,21]. As a result, it is usually necessary to screen a large number of transformants to identify a strain with acceptable levels of transgene expression. However, the fraction of highly expressing transformants can be substantially increased through bicistronic expression of the gene of interest (GOI) and the selectable marker from the same mRNA [23,24]. The first bicistronic system developed for Chlamydomonas uses a viral sequence motif called "2A", which allows two genes encoded as a translational fusion to be produced as separate entities due to a ribosomal skip event at a particular place in the sequence [23,25]. A second system achieves bicistronic expression simply by putting the start codon of the second gene in sequence a mere 6 nucleotides ("6N") behind the stop codon of the first gene [24]. This setup relies on ribosomes to continue scanning along the mRNA and reinitiating translation of the second gene after having terminated translation of the first. Here, the ability of these two systems to drive expression of a Venus fluorescent reporter when using the paromomycin resistance gene *AphVIII* as selectable marker is compared.

When working with chloroplast and mitochondrial genomes, restoration of photosynthesis and respiration respectively through re-introducing the native gene into a mutant background has been successfully used as alternative to antibiotic selectable markers [12,13]. Following a similar strategy, it should be possible to use nuclear genes as functional reporters of restored organelle activity. For the chloroplast, a number of acetate-requiring strains defective in photosynthesis due to a characterized loss-of-function mutation in a nuclear gene are available [5,7]. One well-characterized candidate is the small subunit of Rubisco (genes: *RBCS1* and *RBCS2*, protein: RBCS). Knock-out mutants (*rbcs*) exist that are stably non-photosynthetic [26] and which can be restored by re-introducing *RBCS1* or *RBCS2* into the nuclear genome [26,27].

To use *rbcs* as a tool that not only allows restoration of photo-autotrophy to be selected for, but also subcellular localisation to be tracked, it would be ideal to combine this functional reporter with a fluorescent reporter such as Venus. While several studies have published RBCS-fluorescent reporter fusions [23,28,29], none have used such a construct to rescue an *rbcs* mutant, suggesting that tagging RBCS may interfere with protein function. Rather, fusions were expressed in wild-type strains alongside the native, untagged RBCS, in which case the fusion protein localizes to the pyrenoid [23,28,29]. The pyrenoid is a proteinaceous, liquid-like organelle of functional importance to the algal carbon-concentrating mechanism that is located in the chloroplast stroma and traversed by specialized thylakoid membranes [30–32].

Since the pyrenoid is the place where the vast majority of Rubisco holoenzymes is typically found [33], this correct localization of the fusion hinted that a certain proportion of tagged RBCS may be tolerated.

This study therefore set out to develop a setup where RBCS is produced in an untagged form the majority of the time, but where a large enough fraction of fused RBCS-Venus is produced to allow for localization via fluorescence microscopy. Such expression of multiple versions of a protein with different lengths was achieved by using a leaky stop codon, inspired by viral read-through [34,35]. Here, read-through is demonstrated in Chlamydomonas for the sequence TAG-CAA-TTA, introducing the use of leaky stop codons as a molecular tool for purposes other than to investigate read-through [36–38].

## Results

In order to find a vector that would give high levels of a fluorescent reporter in a large fraction of recovered transformants, current state-of-the art expression systems were compared (Fig 1A). Firstly, as a negative control ('No Venus'), a vector driving the expression of only the selectable marker, paromomycin resistance gene *AphVIII* (AphVIII$^R$), via the same *HSP70/RBCS2* promoter (AR$^P$) and *RBCS2* terminator (R2$^T$) as in the other constructs was used. Secondly, the most common expression system currently used among the Chlamydomonas community is one where the gene of interest (GOI) is physically linked to the selectable marker by virtue of being on the same piece of DNA, but the two genes are driven by independent sets of promoters and terminators [29,39]. Such a 'separate promoters' setup was used employing the fluorescent reporter Venus as GOI, and using a second promoter from *PSAD* (PSAD$^P$). Thirdly, two bicistronic expression systems were compared: a 'ribosomal skip' system where the two genes are linked in a translational fusion via a viral 2A sequence [23,25], and a 'ribosome reinitiation' setup where a stop codon followed by the six nucleotides TAGCAT separates the upstream GOI from the downstream selectable marker [24]. Finally, in the 'extra intron' construct a second *RBCS2* intron was inserted into the Venus coding sequence, with the aim of further increasing expression [14,19,27].

To compare these constructs, a simple read-out of expression levels based on the normalized fluorescence of individual transformant lines in the T222+ wild-type background, quantified in a fluorescence plate reader, was used (Fig 1B). Transformants of the 'no Venus' control plasmid, expressing the selectable marker only, show a small amount of auto-fluorescence at Venus emission wavelengths. Raw fluorescence values of all transformants were normalized to the 'no Venus' population, meaning Venus fluorescence values in Fig 1B are shown as multiples of autofluorescence [24]. Only transformants with fluorescence levels outside a 95% confidence interval defined by the 'no Venus' population are considered as expressing detectable levels of Venus. As expected, the 'separate promoters' construct gave rise to several GOI-expressing transformants, while a majority of transformants have undetectable levels of Venus. In contrast to previous reports [23,25], the 'ribosomal skip' setup failed to give rise to any GOI expressing transformants with our combination of GOI and selectable marker. Meanwhile, a large majority of 'ribosome reinitiation' construct transformants showed expression of the GOI in our hands. Levels of expression could further be increased by introducing RBCS2 intron 2 as 'extra intron' into the vector.

A Western blot shows that normalized fluorescence signal intensity corresponds to Venus expression levels as detected *via* the FLAG-tag (Fig 1C). While no absolute quantification of Venus protein levels was done, plotting a quantification of Western bands against Venus fluorescence (Fig 1D) suggests that a relatively large amount of Venus protein is required to generate a detectable Venus fluorescence signal: a Venus fluorescence value of 1 corresponds to a

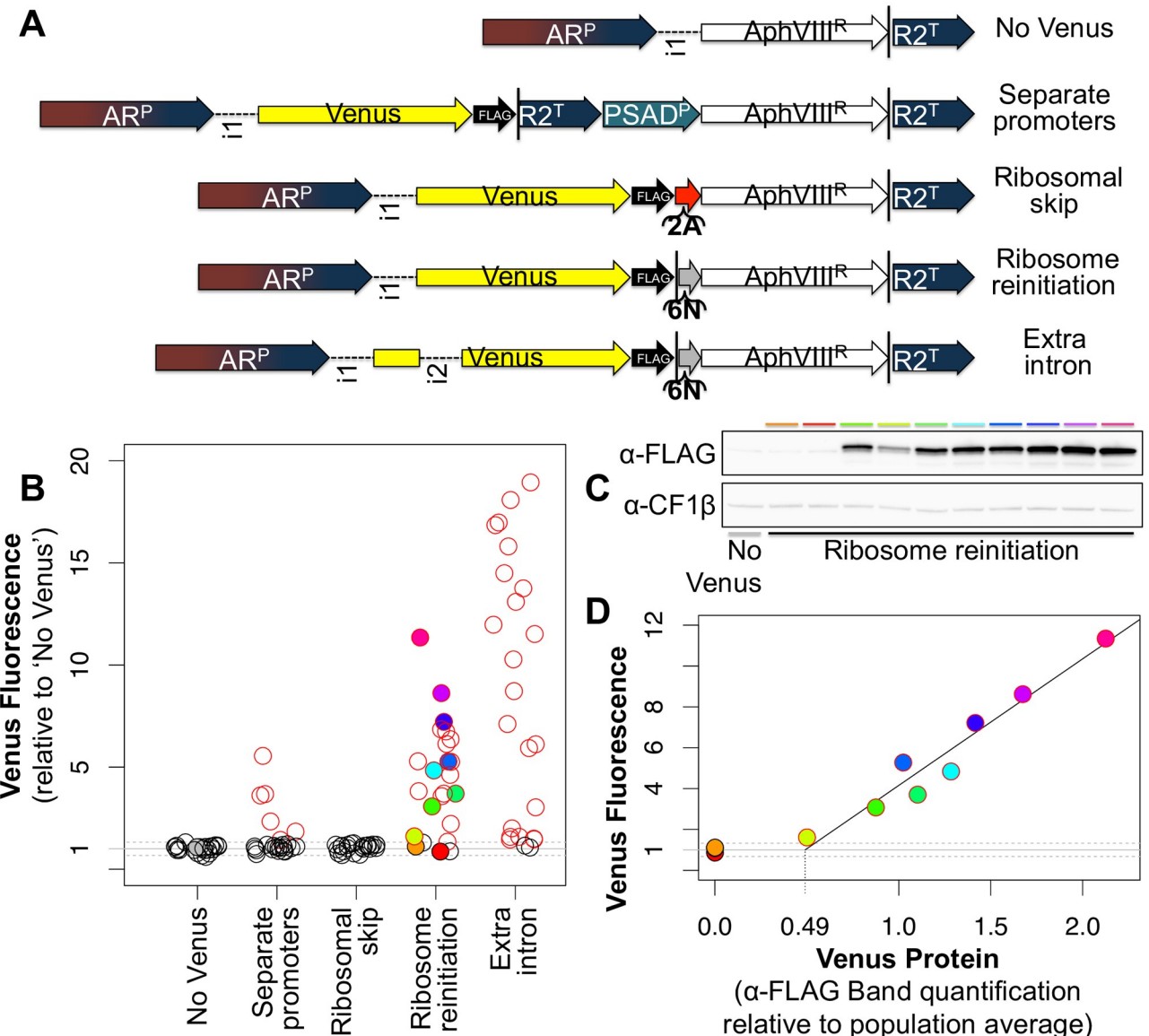

**Fig 1. A high fraction of transformants with high transgene expression can be achieved through ribosome reinitiation combined with multiple introns.** (A) Chlamydomonas expression constructs are composed of the following elements—promoters: chimeric *HSP70-RBCS2* (AR$^P$), *PSAD* (PSAD$^P$); terminator: *RBCS2* (R2$^T$); introns: *RBCS2* 1 (i1) and 2 (i2); protein coding genes: Paromomycin resistance (AphVIII$^R$), Venus fluorescent reporter (Venus); tag: triple-FLAG (FLAG); bicistronic expression elements: Viral 2A sequence (2A), the six nucleotides TAG CAT (6N); STOP codons are indicated by black vertical lines (|). (B) For each construct, fluorescence of 24 independent transformant lines is shown. Venus fluorescence intensities were normalized to autofluorescence as measured in 'No Venus' transformants; red circles indicate transformants that lie outside the 95% confidence interval for 'No Venus' (indicated by dashed lines). Filled circles denote those transformants that were analysed by Western blot (C) as indicated by the colour code. Whole cell protein extracts normalized to 2μg chlorophyll were loaded in each lane. Antibodies against the FLAG-tag at the Venus C-terminus (α-FLAG) and, as loading control, chloroplast ATP-synthase subunit CF1β (α-CF1β) were used. Blots were cropped for clarity, raw images are provided in S1 Raw images. (D) Venus fluorescence levels from (B) are plotted against Venus protein levels, quantified from (C) as Western band intensity relative to the population average. A linear regression line is drawn based on transformants with detectable Venus fluorescence and protein levels (R$^2$ = 0.94). DNA sequences of construct elements and the numeric data underlying the graphs are provided in S1 Data.

Venus-FLAG Western band intensity 0.49 times as high as the population average (or 0.54 to reach the 95% confidence interval). In other words, Venus fluorescence is indistinguishable from autofluorescence in 'ribosome reinitiation' transformants expressing Venus-FLAG up to a level that is ~50% of the population average.

In order to develop a leaky stop codon for Chlamydomonas, three constructs were characterized based on inserting the coding sequence of RBCS2 followed by a linker sequence [29] upstream of the coding sequence of Venus (Fig 2). The first construct (Fig 2A) constitutes a translational fusion of RBCS with downstream Venus through codons GGC ATG CAT. Expression of this construct in *RBCS1* and *RBCS2* double-knockout strains Cal13.1B- or T60 gives rise to a bright fluorescence spot in the Venus channel (in yellow). Comparing Venus to chlorophyll auto-fluorescence (in magenta), which indicates the position of the chloroplast, shows that the RBCS-Venus fusion protein localizes to the canonical pyrenoid location. The pyrenoid is visible as a characteristic dip in chlorophyll auto-fluorescence [29]. In the second construct (Fig 2B), the *RBCS2* coding sequence is followed by the stop codon TAG rendered leaky by the sequence CAA TTA. Expression of this construct also gave rise to Venus fluorescence from the pyrenoid, demonstrating that the leaky stop codon gives rise to a sufficient read-through rate for fluorescence detection. Reduced brightness compared to the fusion construct is as expected if a smaller fraction of expressed RBCS contains Venus. Finally, in a third construct, RBCS was separated from downstream sequences by the stop codon TAA (Fig 2C). In this setup there is no detectable Venus fluorescence. Residual signal in the Venus channel stems from auto-fluorescence that is equally observed in a strain lacking Venus altogether [40].

Comparing the transformation efficiencies when selecting for photo-autotrophy (Fig 2D), it is of note that the 'fusion' construct indeed has a rate close to zero (individual transformants do appear). This result confirms that photosynthetic activity is severely compromised by the translational fusion of RBCS to Venus. By contrast, the leaky stop codon shows transformation efficiencies close to the normal stop codon construct, demonstrating that the majority of RBCS is produced untagged. The leaky stop codon thus works as intended, generating a large enough fraction of untagged RBCS to restore photosynthesis while also producing sufficient amounts of the Venus fusion to allow for fluorescence microscopy. The fact that transformation rates for the leaky stop codon are consistently lower than for the normal stop suggests that the tag does still functionally interfere to some extent, even when only a fraction of RBCS copies carry one.

When selecting for paromomycin resistance (Fig 2E), transformation efficiencies are not only generally higher but also the trend is inversed: now the fusion construct shows the highest rates, in line with bicistronic expression *via* ribosome re-initiation proceeding as normal. The leaky stop construct generally shows somewhat lower values, consistent with only a fraction of translational reads continuing all the way to the end of Venus, which presumably reduces the availability of ribosomes for re-initiation. The normal stop codon shows the lowest values, yet transformation efficiencies even for this construct are far above zero. Being 891bp downstream of the *RBCS2* stop codon, ribosome reinitiation at the *AphVIII* start codon thus appears to occur less frequently, but still often enough to result in between 10–30% of the transformation efficiency of the fusion construct where the distance is only 6 nucleotides.

In a Western blot (Fig 3A), the RBCS-Venus fusion protein could be observed only for 'fusion' construct transformants when using a polyclonal antibody recognizing both RBCL and RBCS (α-Rbc). Unfortunately, a direct quantification of stop codon leakiness from a comparison of tagged and un-tagged protein in the same strain [38]) is thus not possible. The fusion protein could be observed using an antibody against the FLAG-tag at the Venus C-terminus (α-FLAG). A relative quantification of these bands would suggest a read-through rate of ~7.5% (Fig 3A). However, even assuming equal loading, such a direct comparison between individual strains is problematic because random integration of the transgene into the genome results in highly variable expression levels [21,22]. Note that neither RBCS-Venus nor free Venus could be detected in transformants expressing the 'stop' construct.

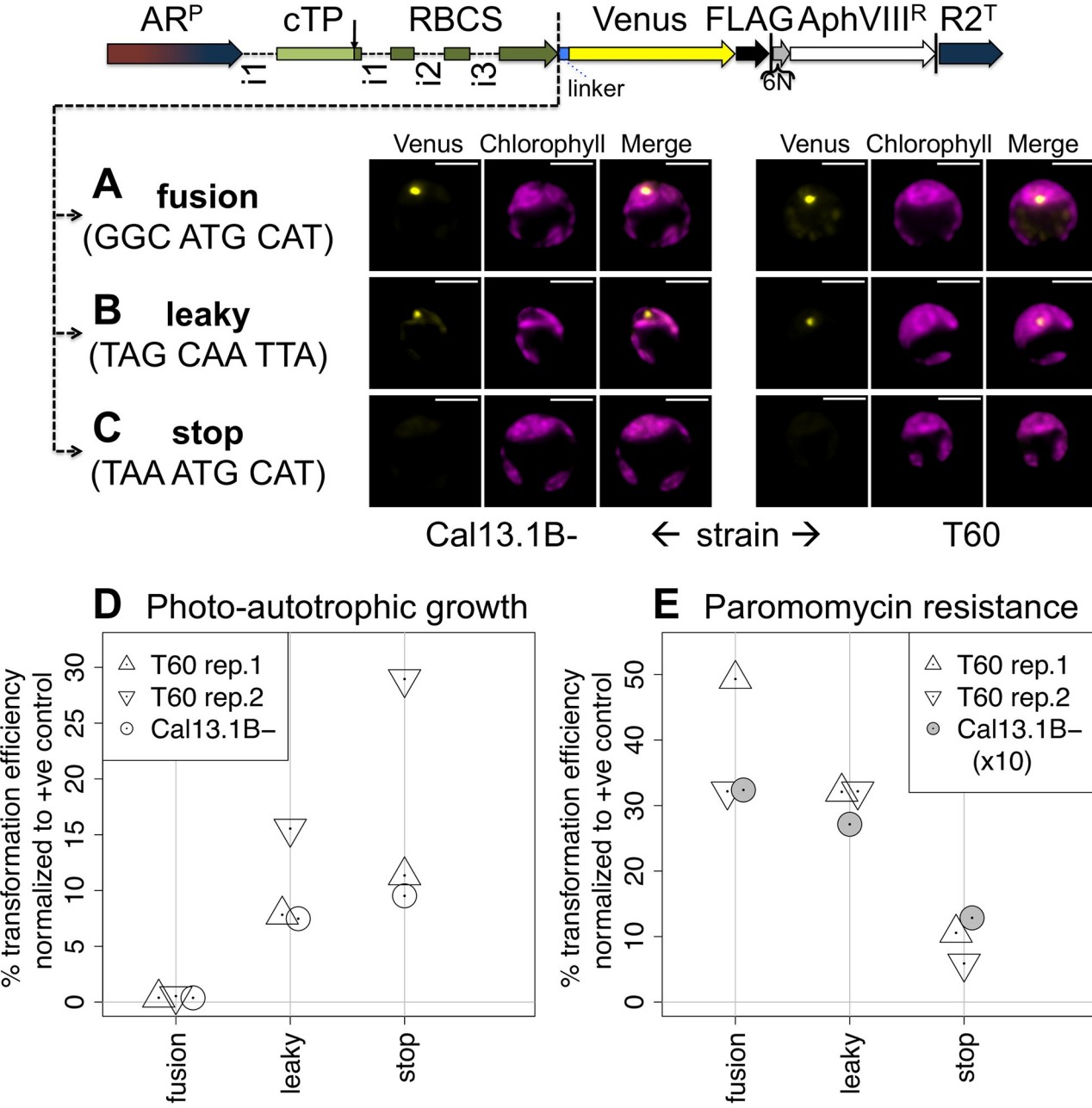

**Fig 2. Use of a leaky stop codon enables photosynthetic rescue using a tagged RBCS.** Constructs were generated by inserting the genomic sequence of Rubisco small subunit 2 (RBCS), including the native targeting peptide (cTP) and introns (i1–i3), followed by a flexible linker sequence (linker), upstream of Venus in the 'ribosome reinitiation' expression vector (see Fig 1). RBCS is separated from the flexible linker by three codons, constituting either (A) a translational **fusion**, (B) a **leaky** stop codon or (C) a normal **stop** codon. DNA sequences are detailed in S1 Data. Typical epifluorescence images of resulting transformants in *RBCS1* and *RBCS2* double-knockout strains Cal13.1B⁻ and T60 show fluorescence from the Venus channel in yellow and chlorophyll autofluorescence in magenta. Scale bars are 5µm. Brightness was adjusted for clarity, original files are available in S2 Data. Transformation efficiencies (D, E) for each construct are shown as percentage relative to a positive control vector encoding RBCS only and selected for photo-autotrophy. Three independent transformation experiments used either T60 (two biological replicates) or Cal13.1B⁻ as host strains. Transformants were selected either for photo-autotrophic growth (D) or paromomycin resistance (E). Values for Cal13.1B- in (E) were divided by 10 to fit on the same scale as T60 values. For comparison: direct colony counts for stop constructs were 29, 54 and 200 in (D) and 27, 11 and ~2700 in (E) for T60 rep.1, T60 rep.2 and Cal13.1B- respectively. Colony counts and calculations are provided in S2 Data, photos of the transformation plates in S3, S4 and S5 Data for T60 rep. 1, T60 rep. 2 and Cal13.1B- respectively.

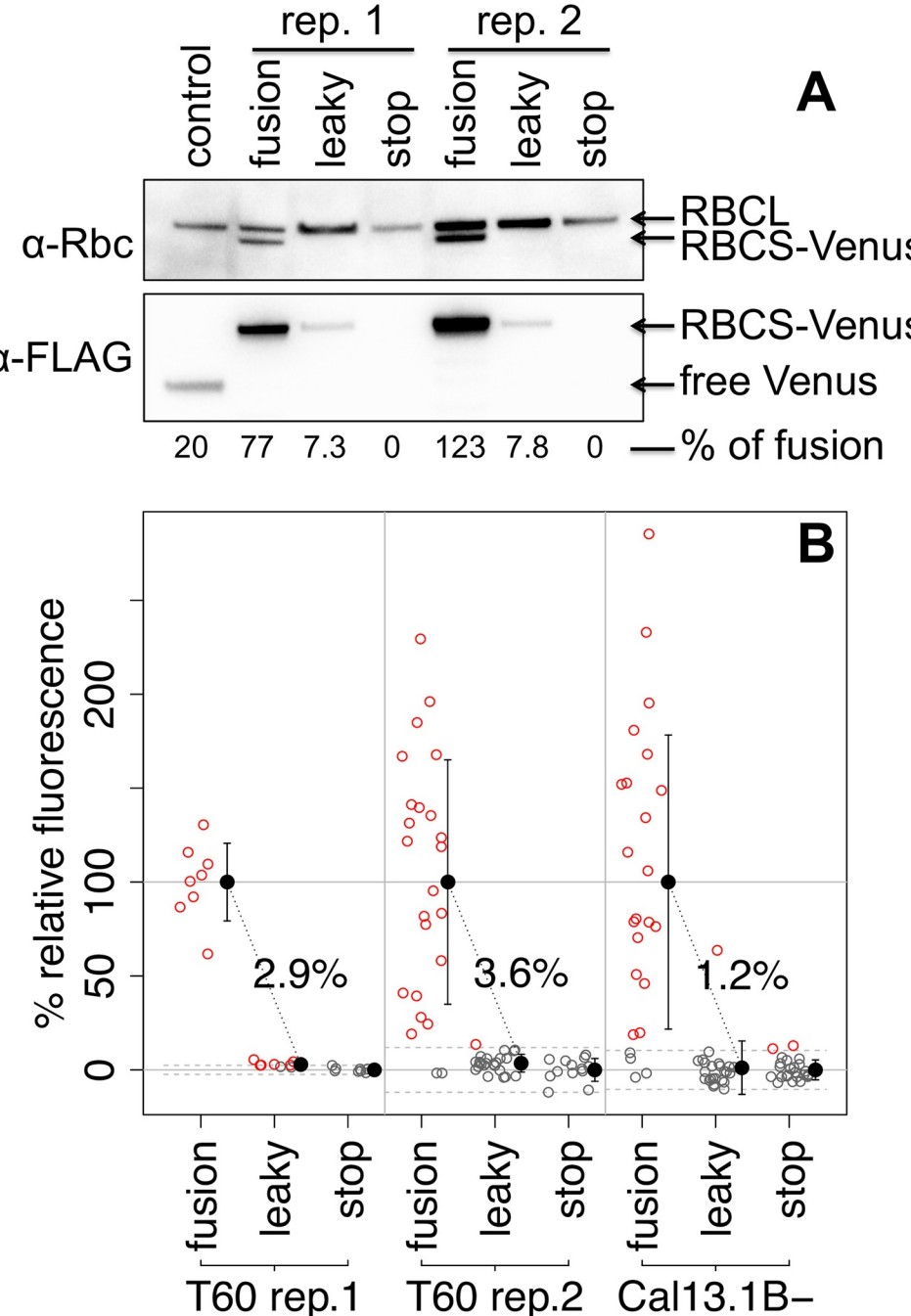

**Fig 3. The leaky stop-codon has read-through rates of ~2.5% (±0.7).** (A) Whole cell protein extracts from transformants of indicated constructs in a RBCS-knockout strain (T60), with loading normalized to 2μg chlorophyll per lane, were immunolabelled with either a polyclonal antibody against Rubisco (α-Rbc) that recognizes both large (RBCL) and small (RBCS) subunits, or an antibody that recognizes the FLAG-tag (α-FLAG) at the Venus C-terminus. A 'ribosome reinitiation' construct transformant was used for the 'control' lane as this strain contains free Venus and no RBCS. Blots were cropped for clarity, raw images are available in S1 Raw images. Bands of the α-FLAG blot were quantified and relative amounts indicated in percent, with 100% corresponding to the average across the two 'fusion' replicates. (B) For each biological replicate, Venus fluorescence of a randomly drawn sample of transformants selected for paromomycin resistance was quantified and plotted after normalization setting the 'stop' construct average as 0 and the 'fusion' construct population average as 100%. Empty circles represent individual transformants, drawn in red if the value exceeds the 95% confidence interval for 'stop' construct transformants indicated by dashed lines, or in grey if not. Population averages and standard deviation are plotted in black to the right of each distribution. Population averages for the 'leaky' construct are written above the point cloud. All numeric data is available in S1 Data.

In order to arrive at a more reliable estimate of stop codon read-through rates in the 'leaky' construct, population-level Venus fluorescence data was considered (Fig 3B). Assuming mRNA expression in individual transformants is influenced only by the genomic locus, any differences in average Venus fluorescence across populations of transformants must be due to differences in Venus translation rates. When normalizing Venus fluorescence values such that the 'stop' construct population average is set to zero and the 'fusion' construct average is set to 100, average Venus fluorescence across 'leaky' construct transformants directly corresponds to stop-codon leakiness. Thus, read-through rates ranged from 1.2% to 3.6% between the three biological replicates (Fig 3B), with a mean of 2.5% and a standard error of ±0.7.

## Discussion

### 2A constructs require abundant selectable marker expression

Comparing different vectors confirmed several aspects governing expression in Chlamydomonas that had previously been described: bicistronic expression results in a higher fraction of GOI expressing transformants [23,24] and addition of introns increases transgene expression levels [19]. In particular, RBCS introns 1 and 2 have been documented to reinforce each other in boosting expression [27]. Expression levels might be increased further by exchanging the terminator for that of FDXI or PSAD, since the RBCS2 terminator used here may be liable to producing an antisense RNA [14,41,42].

However, in contrast to previous reports [23,25], no GOI-expressing transformants could be detected when using the 'ribosomal skip' construct with a viral 2A sequence (Fig 1A and 1B). Promoter, fluorescent reporter gene and selective marker were all changed relative to pQN4 [23], which was used as source of the viral 2A sequence. Changing the selectable marker appears most likely to account for the difference: previously, the 2A sequence has exclusively been used with *ble*, whose gene product relies on high affinity binding to prevent Zeocin and related antibiotics from causing DNA damage [43,44]. Sequestration necessitates the production of large amounts of the *ble* gene product, since each copy of the protein can only bind a single molecule of the antibiotic [45]. Since the 2A-sequence results in quasi-stoichiometric quantities of all proteins in sequence [23], use of *ble* should thus specifically select for transformants with relatively high expression levels.

By contrast, the paromomycin resistance marker AphVIII is an ATP-consuming enzyme with aminoglycoside 3'-phosphotransferase activity [45]. Very low expression levels clearly suffice to confer antibiotic resistance, allowing for relatively high transformation efficiencies seen even in the stop construct (Fig 2C), where ribosome reinitiation at *AphVIII* must be very rare. On the other hand, the enzymatic activity of AphVIII might render overexpression above a certain point deleterious and selected against. Given that a relatively large amount of Venus protein appears to be required to give detectable levels of Venus fluorescence (Fig 1D), selection against high levels of AphVIII could account for the lack of GOI expression seen for the viral 2A 'ribosomal skip' construct in this study (Fig 1B). Meanwhile, ribosome reinitiation results in much lower expression of the downstream gene [24]–in line with the lack of any detectable Venus expression in stop construct transformants (Figs 2C and 3A)–allowing for highly expressed upstream GOIs when using *AphVIII*. It would be interesting to see if the use of *ble* as downstream gene in the context of ribosome reinitiation could be used to specifically select strains with very high GOI expression. Here, *AphVIII* was chosen for all constructs because one of the intended target strains, Cal13.1B-, already carries a *ble* resistance cassette (Table 1) [5,46].

**Table 1. Strains used in this study.**

| Strain | T222+ | Cal13.1B- | | T60 |
|---|---|---|---|---|
| Chlamydomonas resource centre strain ID | CC-5101 | | | CC-4415 |
| Description | Wollman lab wild-type | Niyogi lab *rbcs1,2* | | Spreitzer lab *rbcs1,2* |
| Mating type | + | - | | - |
| Genotype | carries mutations *nit1* and *nit2* | carries a deletion at chromosome 2 pos. 6,908,477–6.943.599, encompassing *RBCS1* and *RBCS2* | | carries a deletion of the *RBCS*-locus at chromosome 2, encompassing *RBCS1* and *RBCS2* |
| | | carries mutations *cw$_d$*, *arg-7-8* (aka. *arg2*) | | carries mutations *nit1* and *nit2* |
| | | expresses transgene *ARG7.8* | | expresses transgene *ble* |
| Phenotype | unable to use nitrate | acetate requiring and high light sensitive; unable to use nitrate; resistant to Zeocin | | acetate requiring and high light sensitive; lacks a cell wall and flagella and cannot be crossed |
| Genealogy | a 137c derivative | progeny of a cross between wild-type S24- (CC-5100) and strain CAL005_01_13 (CC-4691), which was generated by insertional mutagenesis of plasmid pSP124S into strain 4A+ (CC-4051), a 137c derivative | | generated by insertional mutagenesis of plasmid pARG7.8 into strain *cw$_d$ arg-7-8 mt*- (CC-3395) |
| Reference | [57] | [5] | | [26] |

## A leaky stop codon can be used as a molecular tool

Stop codon read-through has been recognized as a mechanism to generate multiple variants of a protein from a single gene in viruses for some time [34,37,47]. Like all biological processes, translation termination is associated with a certain error rate, which is not identical between the three stop codons in the standard code. This error rate can further be influenced by RNA secondary structures, or simply the immediate sequence environment of the stop codon, meaning that certain sequence constellations can lead to significant read-through rates on the order of 1–10% [34,48–50]. In eukaryotes, leaky stop codons appear to be generally selected against [51,52], although examples where read-through fulfills a regulatory function have been documented [53,54]. A fair amount of research has gone into exploiting leakiness therapeutically in cases where disease is caused by premature stop codons [55,56]. Perhaps surprisingly though, leaky stop codons had not yet found their way into the molecular biology tool-box, except for the purpose of investigating read-through [36–38,48].

As this study demonstrates, functional read-through can provide a solution to the long-standing problem that reporter gene fusions sometimes impede the function of a gene of interest, exemplified here by the failure of the RBCS-Venus fusion construct to restore photo-autotrophy (Fig 2D). Since it was not clear to what extend RBCS may be able to tolerate any C-terminal sequence modifications, 3' sequence elements were used to generate stop codon read-through, which are anyway thought to be the main determinants of leakiness [34,37]. The sequence TAG-CAA-TTA in particular was chosen because its efficacy had been demonstrated in *S. cerevisiae* as well as tobacco [36,37], suggesting eukaryotes across a phylogenetic range that includes Chlamydomonas may be susceptible. With an estimated read-through rate of ~2.5% (Fig 3B), the leakiness generated by this sequence in Chlamydomonas is much lower than the 30% reported in yeast [36], but on par with the ~3% reported for mammalian tissue culture [48] and the 2–5% observed in TMV-infected tobacco [49]. Importantly, this rate is sufficient to visualize the fluorescent reporter under the microscope, while not interfering with the functional rescue of photo-autotrophy (Fig 2D).

## Materials and methods

### Generation of constructs

Constructs were Gibson assembled as described in [40] based on pMO611, a variant of pMO449 [24] where an upstream start codon was mutated to CTG. The 'ribosome reinitiation' and 'separate promoters' constructs are plasmids pMO449 and pMO518 respectively [24]. The 'no Venus' construct was generated by deleting Venus-FLAG-TAGCAT from pMO611, for the 'ribosomomal skip' construct the stop-TAGCAT sequence was replaced by the 2A sequence from pQN4 [23], and RBCS2 intron 2 in the 'extra intron' construct was amplified from pLM005_RBCS [28] which also served as template for RBCS and linker sequences for the 'fusion', 'leaky' and 'stop' constructs in Fig 2. Constructs were verified by sequencing (Eurofins). DNA sequences of all elements are provided in S1 Data.

### Strains, transformation and culture conditions

Linear transformation cassettes were cut out of circular plasmids using EcoRV-HF (NEB) and transformed into wild-type strain T222+ or *RBCS1* and *RBCS2* double-knockout strains Cal13.1B- or T60 (Table 1). Transformation *via* electroporation and screening for expression of the fluorescent reporter in a fluorescence plate reader (CLARIOstar, BMG labtech) were performed as described in [24], except that 4μl of DNA at 2μg ml$^{-1}$ was used. Transformants were selected either for antibiotic resistance on Tris-Acetate-Phosphate (TAP) plates containing paromomycin in the dark, or for photo-autotrophy on Minimal-plates in constant 30–50μmol photons m$^{-2}$ s$^{-2}$. For Western blots, fresh cell material was scraped from TAP-paromomycin plates; for microscopy, cells that had been grown in TAP in 96-well plates for screening were used.

### Fluorescence distributions and transformation efficiencies

Distributions of Venus fluorescence (Figs 1B, 1D and 3B) across transformants are derived from fluorescence plate reader data as follows [24]. Absorbance at 750nm (OD$_{750}$), Venus fluorescence (excitation: 490–505 nm, emission: 530–550 nm, Gain = 1500) and chlorophyll autofluorescence (excitation: 465–480 nm, emission: 670–690 nm, Gain = 1500) values recorded in wells not containing algae were substracted from values recorded in sample wells. Zeroed Venus fluorescence values were further divided by zeroed absorbance (Fig 1B and 1D) or zeroed chlorophyll (Fig 3B) values to account for different cell concentrations, and normalized to the mean of a population of transformants expressing the "No Venus" control construct that was transformed and screened alongside samples. To evaluate leakiness (Fig 3B), for each biological replicate, the mean of a population of transformants expressing the 'stop' construct was substracted from sample values, which were subsequently divided by the mean across 'fusion' construct transformants. Transformation efficiencies were calculated by counting transformant colonies and then dividing that number by the number of cells used for transformation. Transformation efficiencies were further normalized to those of a construct where the expression of the rbcs2 genomic sequence is driven by the rbcs-hsp70 promoter and the rbcs2 terminator and where transformants were selected for photoautotrophy. Calculations and plotting were done in R (version 3.6.1). Original measurements and derived data are provided in S1 Data, photos of the transformation plates in S3, S4 and S5 Data for T60 rep. 1, T60 rep. 2 and Cal13.1B- respectively.

### Western blots and microscopy

For Western Blots, whole cells were boiled for 50 seconds in an SDS/DTT buffer (0.1M DTT/ Na$_2$CO$_3$, 10mM NaF, 2% SDS, 12% Sucrose, 1x Roche complete Mini proteinase inhibitor

cocktail). Cell debris was discarded by centrifugation (tabletop centrifuge, maximum speed, 15min, 4°C). The chlorophyll concentration in the crude whole cell protein extract was estimated by diluting a 5μl aliquot in 1ml distilled water and recording the absorbance at 680nm ($OD_{680}$), corrected by any absorbance at 770nm ($OD_{770}$), as follows: $1\mu g = chl/\mu l = 0.11$ ($OD_{680} - OD_{770}$). Gels and blots were performed as described in [40] using a precast gel (Bio-Rad) and a 0.1μm nitrocellulose membrane that was cut horizontally and treated with primary antibodies α-FLAG (monoclonal, mouse, Sigma Aldrich F1804, dilution: 1:10 000), CF1-β (polyclonal, rabbit, raised against CF1-β [58], dilution: 1:50 000) or Rubisco (polyclonal, rabbit, raised against Rubisco holoenzyme [59], dilution: 1:20 000). Horseradish peroxidase conjugated anti-mouse and anti-rabbit secondary antibodies (Promega) were revealed using Clarity Western ECL reagents (Bio-Rad) on a ChemiDoc Touch Imaging System (Bio-Rad). Bands were quantified using Image Lab software (version 6.0.0, Bio-Rad). Blots were cropped for clarity, and figures prepared in Powerpoint (Microsoft). Raw images are provided in S1 Raw images, quantification measurements in S1 Data.

Microscopy was performed as described in [40]. In Fiji (http://fiji.sc/Fiji, version 2.0.0-rc-69/1.52p), individual cells were cropped out of larger fields-of-view, brightness was adjusted for clarity linearly and uniformly across each image, and linear yellow or magenta look-up tables were applied. Original image files are available in S2 Data. Final figures were assembled in Powerpoint.

## Supporting information

**S1 Raw images. Raw image files for Western blots.**
(PDF)

**S1 Data. Tabulated numeric and sequence data.**
(XLSX)

**S2 Data. Original fluorescence microscopy image files.**
(ZIP)

**S3 Data. Photos of transformation plates for T60 rep. 1.**
(ZIP)

**S4 Data. Photos of transformation plates for T60 rep. 2.**
(ZIP)

**S5 Data. Photos of transformation plates for Cal13.1B.**
(ZIP)

## Acknowledgments

I would like to thank Francis-André Wollman and Michael Schroda for critical reading of the manuscript, Richard Kuras and Yves Choquet for helpful discussions and advice, and David Herrin for his helpful public review.

## Author Contributions

**Conceptualization:** Oliver D. Caspari.

**Data curation:** Oliver D. Caspari.

**Formal analysis:** Oliver D. Caspari.

**Investigation:** Oliver D. Caspari.

**Methodology:** Oliver D. Caspari.

**Software:** Oliver D. Caspari.

**Validation:** Oliver D. Caspari.

**Visualization:** Oliver D. Caspari.

**Writing – original draft:** Oliver D. Caspari.

**Writing – review & editing:** Oliver D. Caspari.

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
