## [Decision Letter · Decision Letter 0]

11 Jun 2020

PONE-D-20-11123

Introduction of a leaky stop codon as molecular tool in *Chlamydomonas reinhardtii*

PLOS ONE

Dear Dr. Caspari,

Thank you for submitting your manuscript to PLOS ONE. After careful consideration, we feel that it has merit but does not fully meet PLOS ONE’s publication criteria as it currently stands. Therefore, we invite you to submit a revised version of the manuscript that addresses the points raised during the review process.

We look forward to receiving your revised manuscript.

Kind regards,

Andrew Webber

Academic Editor

PLOS ONE

Journal Requirements:

Additional Editor Comments (if provided):

The review of the manuscript was quite positive. There are a number os suggestions that are worthy of consideration. Hopefully, the authors may already have some data to improve some figures. If not, revising the discussion to account for concerns would be very helpful. Please provide details of all changes with the revision.

Reviewers' comments:

Reviewer's Responses to Questions

**Comments to the Author**

1. Is the manuscript technically sound, and do the data support the conclusions?

Reviewer #1: Yes

2. Has the statistical analysis been performed appropriately and rigorously? 

Reviewer #1: Yes

3. Have the authors made all data underlying the findings in their manuscript fully available?

Reviewer #1: Yes

4. Is the manuscript presented in an intelligible fashion and written in standard English?

Reviewer #1: Yes

5. Review Comments to the Author

Reviewer #1: The author has provided a potentially new tool for nuclear gene engineering in Chlamydomonas by demonstrating the use of a leaky stop codon that allows bicistronic expression of a selectable marker and another gene from the same promoter. Such a system greatly facilitates nuclear engineering in Chlamy. Thus, I think that a number of research groups will find this work useful. However, I have a few suggestions/comments that may improve the clarity of the protein data, and manuscript.

1. The comparison of tagged Venus expression to CF1 Beta by immunoblotting in Fig. 1D, and in the attendant text (lines 185-190), will be confusing and possibly misleading for some readers. It is not really valid to compare in any relative-quantitative way two different antigen-antibody interactions, and Fig. 1D could be interpreted to mean that Venus levels are much greater than CF1 Beta levels, which is probably not true, right?

CF1 Beta levels detected with an antibody may be useful as a loading control (which means using them internally to adjust the “real” protein loads), although in our hands immunodetection of very abundant proteins (like CF1 subunits) with sensitive chemiluminescence methods is not very accurate unless the protein load is very, very low. Protein stain is probably better for revealing protein loads if you are loading a lot of cellular protein (say, 10 or 20 micrograms of protein in a mini-gel lane), and your control immunotarget is very abundant.

Incidentally, the detection system should be briefly described as well as the amounts of protein loaded in each of the gel lanes; the readers should not have to read another paper or consult their Ouija boards to find those things out.

Finally, if you want to say that a lot of Venus needs to accumulate before you can see the fluorescence, then you should quantify Venus in an absolute way using quantitative immunoblotting and a known amount of Venus standard on the gels.

2. The Figure 3A western also has some confusion. What is the control lane? What are the protein loads, and what is equalized? Finally, I don’t understand the “% of fusion” quantification below the blots; shouldn’t one of them be 100%? It might be clearer that way.

3. I suggest that a Table of the host strains with their genotype/characteristics might be helpful. Along that line, why does strain Cal13 transform much better than T60?

6. PLOS authors have the option to publish the peer review history of their article (what does this mean?). If published, this will include your full peer review and any attached files.

Reviewer #1: Yes: DAVID L. HERRIN

---

## [Author Response · Author response to Decision Letter 0]

19 Jun 2020

I thank David L Herrin for his helpful public review and am glad to hear he thinks people in the community will find my work useful.

Point 1: (Figure 1D and corresponding legend and text)

I agree with him that protein abundances should not be compared directly when inferred from different antibodies without absolute quantification. Most certainly I would expect there to be more CF1 beta than Venus. My use of CF1 beta in Fig. 1D was indeed as a loading control, and I simply used the resulting values directly as an arbitrary unit scale, if you will, to plot Venus quantities on. However, I must admit that this can very easily be misunderstood, and I have thus changed the scale to units relative to the average across my Venus samples. As I did refer to quantities of Venus relative to the population in the text anyway (lines 185-190), this should make the paper easier to understand. Supplementary Table S2 has been updated accordingly.

Indeed the reader should not need Ouija boards or tarot cards, I apologize for the oversight of not having indicated loaded quantities. Crude protein extracts were normalized to 2µg chlorophyll in each lane, which I have now specified in figure legends (lines 158, 271). A description of how bands were revealed was added in the methods section (lines 431-434).

An absolute quantification of Venus would certainly strengthen the case for saying a lot of protein is needed to generate detectable fluorescence. However, I do think the relative quantification I present is suggestive enough in its own right to state the observation. I have weakened the statement in the text to reflect the ambiguity that remains in the absence of absolute quantification (lines 183-190, line 325). 

Point 2: (Figure 3A and corresponding legend and text)

Again apologies for the oversight, I have now added a clear statement about the identity of the control lane in the legend (lines 273-274); it is a ‘ribosome reinitiation’ construct transformant. 

I understand that the relative quantification of Bands is confusing in that none of the lanes correspond to 100%. However, upon reflection I have decided to keep it that way. This is because 100% is meant to be the ‘real’ protein abundance in ‘fusion’ construct transformants. In the figure, I show two different ‘fusion’ construct transformants, which vary a lot in Venus abundance. If I were to set either of the two replicates to 100%, then I would get an arbitrarily high or low apparent abundance for the ‘leaky’ construct transformants. I do not think it makes sense to treat ‘leaky’ rep1 relative to ‘fusion’ rep1 and ‘leaky’ rep2 relative to ‘fusion’ rep2 either, since there is no a priori reason why they should form sets. Rather, differences in expression levels should be down to differences in the genomic locus that the transgene construct has integrated in. Thus the best thing to do, in my mind, is to take the average of the two ‘fusion’ transformants and set that value to 100%. This is what I have done, and I have updated the legend in a way that hopefully makes this clearer (lines 275-276). 

Point 3: (Table of strains)

Adding a table of strains seems a very good idea to me indeed, and I have done so (line 381, text calls: line 335, 385). I must say that I do not really know why there is such a difference in transformation efficiency between Cal13.1B- and T60. However, I would say that Cal13.1B- is quite similar in that respect to T222+, so I’d argue it is T60 that is the outlier with a very low transformation efficiency. T60 is the most cell wall deficient strain I have ever worked with, and the cells are really quite fragile; my guess would be that this is the reason for my difficulties in transforming it. 

The fact that Cal13.1B- is not so dissimilar to T60 when it comes to strains recovered photo-autotrophically may reflect an observation made by colleagues in the lab, that Cal13.1B- appears to have depressed levels of PSII subunit D1 in addition to the loss of RBCS1,2. However, while my colleagues are currently preparing a publication on a characterization of this strain, I feel that this is beyond the scope of my article and I have therefore refrained from speculating in that direction in the text.

---

## [Editor Report · Decision Letter 1]

27 Jul 2020

Introduction of a leaky stop codon as molecular tool in *Chlamydomonas reinhardtii*

PONE-D-20-11123R1

Dear Dr. Caspari,

We’re pleased to inform you that your manuscript has been judged scientifically suitable for publication and will be formally accepted for publication once it meets all outstanding technical requirements.

Kind regards,

Andrew Webber

Academic Editor

PLOS ONE
---

## [Editor Report · Acceptance letter]

4 Aug 2020

PONE-D-20-11123R1 

Introduction of a leaky stop codon as molecular tool in *Chlamydomonas reinhardtii*

Dear Dr. Caspari:

I'm pleased to inform you that your manuscript has been deemed suitable for publication in PLOS ONE. Congratulations! Your manuscript is now with our production department. 

Kind regards, 

on behalf of

Dr. Andrew Webber 

Academic Editor

PLOS ONE